# Evaluation of the Effectiveness of Medical-Grade Honey and Hypericum Perforatum Ointment on Second-Intention Healing of Full-Thickness Skin Wounds in Cats

**DOI:** 10.3390/ani14010036

**Published:** 2023-12-21

**Authors:** Kyriakos Chatzimisios, Vassiliki Tsioli, Georgia D. Brellou, Emmanouela P. Apostolopoulou, Vasileia Angelou, Emmanouil D. Pratsinakis, Niels A. J. Cremers, Lysimachos G. Papazoglou

**Affiliations:** 1Unit of Surgery and Obstetrics, Companion Animal Clinic, School of Veterinary Medicine, Faculty of HealthSciences, Aristotle University of Thessaloniki, 54627 Thessaloniki, Greece; kchatzimi@vet.auth.gr (K.C.);; 2Clinic of Surgery, School of Veterinary Medicine, University of Thessaly, 224 Trikalon Street, Box Office 199, 43100 Karditsa, Greece; 3Laboratory of Pathology, School of Veterinary Medicine, Faculty of Health Sciences, Aristotle University of Thessaloniki, 54124 Thessaloniki, Greece; mprellou@vet.auth.gr (G.D.B.); emmaapos@vet.auth.gr (E.P.A.); 4Laboratory of Agronomy, School of Agriculture, Faculty of Agriculture, Forestry and Natural Environment, Aristotle University of Thessaloniki, 54124 Thessaloniki, Greece; epratsina@agro.auth.gr; 5Department of Gynecology and Obstetrics, Maastricht University Medical Center, 6202 AZ Maastricht, The Netherlands; niels@mesitran.com; 6Triticum Exploitatie BV/Theomanufacturing BV, Sleperweg 44, 6222 NK Maastricht, The Netherlands

**Keywords:** cat, honey, Hypericum, wound healing

## Abstract

**Simple Summary:**

Medical-grade honey was found to promote cutaneous wound healing in horses, but no effect was reported in dogs. The effect of medical-grade honey or Hypericum on cutaneous wound healing in cats is unknown since there are no controlled studies in this species. Therefore, the objective of this study was to evaluate the effect of the local application of medical-grade honey or Hypericum-based ointment in second-intention healing of cutaneous defects in healthy cats versus their untreated controls and to compare the effectiveness of medical-grade honey and Hypericum. Tissue perfusion was better in wounds treated with medical-grade honey and Hypericum rather than in the untreated controls. Wounds treated with medical-grade honey showed lower edema, higher angiogenesis, and an increased fibroblast concentration compared to Hypericum-treated wounds. Topical application of medical-grade honey or Hypericum did not accelerate the healing process of feline cutaneous wounds.

**Abstract:**

This study aimed to determine the effects of two topical treatments on second-intention wound healing in cats. Eight 2 × 2 cm full-thickness wounds were created, four on each side of the dorsal midline of eight laboratory cats, to receive either medical-grade honey ointment (MGH) and its control (HC), or Hypericum-based ointment (HP) and its control (HPC). MGH or HP ointment was applied to four wounds on the same side, while the remaining four were used as controls, chosen at random. Planimetry, laser Doppler flowmetry, daily physical examinations, and histologic examinations on days 0, 7, 14, and 25 were used to assess the healing of wounds. Tissue perfusion was better in the MGH-treated (2.14 ± 0.18 mm/s) and HP-treated wounds (2.02 ± 0.13 mm/s) than in the untreated controls HC (1.59 ± 0.11 mm/s) and HPC (1.60 ± 0.05 mm/s), respectively (*p* = 0.001). Histopathology revealed that the median edema score was lower in the MGH-treated (2; range 1–4) compared to the HC-treated wounds (3; range 2–4) on day 7 (*p* < 0.05). The median angiogenesis score was higher on day 7 in the MGH-treated (2; range 1–3) compared to the HP-treated wounds (2; range 1–2) (*p* = 0.046). The fibroblast concentration was increased in the MGH-treated wounds (3.5; range 3–4) compared to the HP-treated wounds (3; range 2–4) on day 25 (*p* = 0.046). MGH and HP increased tissue perfusion compared to the untreated controls. The MGH-treated wounds had histologic parameters superior to the HP-treated wounds regarding angiogenesis and fibroblast concentration in cutaneous wound healing in cats. Topical application of MGH and HP did not accelerate the healing process of feline cutaneous wounds.

## 1. Introduction

Cutaneous defects in cats are usually created by surgical excision of tumors, trauma, or infections. The size and location of the wound and various patient factors (i.e., looseness of the skin, radiation therapy, systemic disease, etc.) may influence the decision to allow healing by second intention. There are significant differences in cutaneous wound healing between feline and canine patients, with cats showing delayed healing compared to dogs [1,2]. More specifically, the formation of granulation tissue is slower in cats, and granulation tissue begins to form from the edges of the defect and proceeds to the center of the wound, in contrast to dogs, where the granulation tissue is formed simultaneously on the entire surface of the wound [1]. Studies have shown that subcutaneous tissue in the cat promotes the formation of granulation tissue and the second-intention healing of wounds. Defects with extensive subcutaneous tissue debridement in cats showed a significant delay in healing compared to dogs [2]. A phenomenon known as pseudo-healing has been seen in cats; wounds denuded of subcutaneous tissue may be prone to dehiscence following suture removal [1]. A better insight into the healing mechanism of felines and local wound care products is warranted to improve healing [3]. Studies have been performed comparing many wound care products in second-intention healing in canines [4,5,6,7] and very few in feline patients [8,9,10].

Honey has been used in medicine for several millennia, but due to the discovery of antibiotics, its use has declined [11]. The increase in antibiotic resistance development has led to a resurgence in the use of medical-grade honey. The salutary properties of medical-grade honey are based on its antimicrobial and pro-healing activities, and therefore, it is typically used to advance topical cutaneous wound healing. Medical-grade honey follows strict guidelines and criteria to guarantee its safe and effective use for medical applications [12,13]. For example, medical-grade honey adheres to certain physicochemical parameters, it is not contaminated with potential pollutants, such as herbicides, pesticides, heavy metals, and antibiotics, and it is gamma-irradiated to kill possible endospores that can be present [14]. Different CE-approved and FDA-certified medical-grade honey formulations exist, of which L-Mesitran was the first on the market accredited by both organizations [14]. The antimicrobial activity of medical-grade honey is based on multiple mechanisms, such as its osmotic activity, its acidic pH, and the presence of antimicrobial molecules, such as flavonoids and phenolic compounds. Furthermore, a division can be made between hydrogen peroxide-based honey (e.g., L-Mesitran) and methylglyoxal- and non-peroxide-based honey (e.g., manuka-based Medihoney) [12]. Many products contain manuka honey because this type of honey has been extensively investigated [15]. However, more recent studies have shown that peroxide-based honey can have similar or even stronger antimicrobial activity than non-peroxide-based honey [16,17,18,19]. L-Mesitran contains 40% medical-grade honey and is supplemented with ingredients such as vitamins C and E that enhance its activity and improve its applicability, while Medihoney honey contains 80% manuka honey [20]. L-Mesitran showed more consistent and significantly stronger antimicrobial activity, despite containing half the amount of honey in its formulation [14,16,18]. This has been ascribed to the synergistic activity of the different supplements in its formulation [16,18,21,22,23]. Application of medical-grade honey in human wounds promotes granulation tissue formation and epithelialization, by accelerating the maturation of collagen and converting the pH to acidic, thus releasing free oxygen radicals from hemoglobin molecules, stimulating fibroblast activity, and the proliferation of B and T lymphocytes and macrophages [24,25,26]. As honey is a natural product, its effectiveness may vary depending on the source of the product and its processing methods [27,28]. The reported effects of medical-grade honey on wound healing in animals are controversial. Experimental studies on contaminated cutaneous wounds in horses using manuka honey have shown accelerated healing compared to controls [29,30]. In two prospective randomized clinical studies, local subcutaneous application of L-Mesitran honey improved the healing of cutaneous lacerations in horses and decreased the prevalence of incisional infection in colic surgery in horses compared to controls [31,32]. However, a recent prospective, controlled, randomized, experimental study using manuka honey on acute cutaneous defects in 10 healthy dogs failed to provide evidence to support the application of manuka honey [33]. There is a paucity of publications evaluating the effect of medical-grade honey in cats [8,9].

Hypericum (*Hypericum perforatum*) is the extract of the respective plant, contains flavonoids and other substances, and has been found to promote wound healing and growth of granulation tissue in humans, cattle, and mice [34,35,36,37]. St. John’s Wort is the common denomination of Hypericum perforatum, and its anti-inflammatory and anticarcinogenic action is probably due to the inhibition of 5-lipoxygenase and the production of prostaglandin E2 (PGE2). There is a report of six horses with skin lesions treated successfully with the topical application of a Hypericum perfoliatum-based ointment [38]. Hypermix^®^ is a plant product based on Hypericum and Neem for external use. Hypericum flowers come from certified organic farming. An oily extraction is performed on Hypericum flowers using olive oil as a solvent. The Neem oil used in the production of Hypermix^®^ is extracted from cold-pressed seeds of the Melia Azadirachta Indica plant and subsequently filtered to remove impurities. Hypericum has been also reported to aid in the healing of ulcerated mammary fibro adenomatous changes in a small number of cats [39]. However, no reports evaluating the effect of Hypericum on feline cutaneous wound healing have been published so far. 

To the authors’ knowledge, there are no controlled studies performed on cats to evaluate the effect of medical-grade honey or Hypericum in cutaneous wound healing. Therefore, the objective of this study was to evaluate the effect of the topical application of L-Mesitran soft ointment and Hypermix ointment in second-intention healing of cutaneous defects in healthy cats versus untreated controls and to compare the effectiveness of medical-grade honey and Hypericum. We hypothesized that the use of medical-grade honey and Hypericum might promote second-intention healing of cutaneous defects in cats. More specifically, that both treatment groups would differ from their untreated controls regarding wound area, tissue perfusion, epithelialization, contraction, total wound healing, inflammatory cell infiltration, edema angiogenesis, fibroblast concentration, and collagen production score. Furthermore, that no differences would be observed between the treatment groups regarding wound area, tissue perfusion, epithelialization, contraction, total wound healing, inflammatory cell infiltration, edema, angiogenesis, fibroblast concentration, and collagen production score.

## 2. Materials and Methods

### 2.1. Cats

All procedures were approved by the Research Ethics Committee of Bioethics and Research Deontology of the School of Veterinary Medicine (license number: 633,001/3372) and the State Veterinary Services (license number: 598/ 5 August 2019) in compliance with the laws of the European Union pertaining to the care and humane treatment of laboratory animals. The Animal Welfare Act and the NRC Guide for the Care and Use of Laboratory Animals were followed when using laboratory animals in this study. A power analysis was conducted to enable the use of the fewest number of animals required to accomplish the scientific objectives in an ethically acceptable study.

Eight female spayed fully vaccinated purpose-bred healthy domestic shorthaired cats of median age 2.5 years and median weight 3.5 kg, were included in the study. Health status was based on physical examination, hematology, serum biochemistry analysis, and fecal parasitology. Feline immunodeficiency virus and feline leukemia virus tests were also performed. 

The animals were kept in separate large indoor cages at the Clinic of Companion Animals. Commercial dry maintenance diets and water were freely offered. To acclimate the cats to the experimental protocol, 1 week before the initiation of the study a padded circumferential body bandage, that extended from the cranial thoracic to the caudal lumbar region, was applied on each animal. After completion of the study, all cats were given for adoption [26]. 

### 2.2. Anesthesia

Anesthetic management was performed as described by Angelou et al. (2022) [10]. On day 0, premedication was performed by the administration of acepromazine maleate (Acepromazine; Alfasan, Woerden, The Netherlands) at a dose of 0.05 mg/kg IM and buprenorphine (Bupredine multidose; Dechra Academy, Oudewater, The Netherlands) at a dose of 0.1 mg/kg IM. Anesthetic induction was performed using propofol (Propofol MCT/LCT; Fresenius Kabi Hellas, Athens, Greece) at a dose of 2 mg/kg IV and maintenance was achieved with isoflurane (Isoflurane-Vet; Merial, Milano, Italy) in oxygen after endotracheal intubation. Lidocaine (Xylocaine pump spray; AstraZeneca, Osaka, Japan) was sprayed on laryngopharyngeal structures prior to endotracheal intubation. Lactated Ringer’s solution (LR’s; Vioser, Trikala, Greece) was administered IV at 5 mL/kg/h during the procedure. 

### 2.3. Skin Wound Creation

Each animal was positioned in sternal recumbency and the dorsolateral area from the cranial aspect of the thorax to the lumbosacral junction was prepared for aseptic surgery. Using a sterile skin marker and a millimeter ruler eight 2 × 2 cm squares were designed, four on either side of the dorsal midline 3 cm away from each other, and 3 cm away from the dorsal midline (Figure 1).

The most cranial square was placed just caudally to the caudal border of the scapula and the most caudal square was placed cranially to the iliac crest. The 4 squares on each side were randomized using computer software (random number generator) to receive either treatment or control. Full-thickness skin wounds were created following the removal of cutaneous trunci muscle and subcutaneous tissue using a # 15 blade until the thoracolumbar fascia was visible. At each defect’s corners, 3-0 nylon tacking sutures were placed to prevent the overlying skin from shifting or sliding over the underlying fascia [1]. Bleeding was negligible in most of the wounds, and otherwise it was easily stopped with sterile gauze before the application of the wound care product. On each cat, 2 of the wounds were treated with medical-grade honey ointment (MGH) (L-Mesitran Soft, Triticum, The Netherlands), 2 were treated with Hypericum-based ointment (HP) (Hypermixvet gel, RI.MOS, Mirandola, Modena, Italy), 2 were used as untreated controls for medical-grade honey (HC), and 2 were used as untreated controls for Hypericum-based ointment (HPC). Moving from cranial to caudal wounds, one side received MGH (2 wounds) and HP (2 wounds) ointments and the wounds of the other side served as respective controls. The control groups did not receive ointment. The treatments were applied once daily in an aseptic fashion for 25 days. Following application instructions, both products were applied daily as a thin layer (www.mesitran.com/products/, www.hypermix.it/en/what-is/how-to-use accessed on 20 November 2020). A layer (2–5 mm) of L-Mesitran Soft or a volume of 1 mL Hypermix^®^ was applied to the wounds extending to the perilesional margins, since the instructions for use do not specify an exact quantity per wound area. 

On days 0, 7, 14, and 25, the two cranial wounds were used for planimetry, laser Doppler flowmetry (LDF) measurements, and biopsies designated for histologic assessment. On days 7 and 14, the two caudal wounds were used for biopsies designated for histologic assessment.

### 2.4. Post-Operative Care

Post-operatively, the wounds were covered with two nonadherent dressings with adhesive edges (Cosmopor, Hartmann, Germany), one on each side, a cotton roll pad (Rolta soft padding bandage, Hartmann, Germany) around the trunk, crossing around the forelimbs, and an adhesive bandage (Peha-haft, Hartmann, Germany) placed around the trunk. An Elizabeth collar was applied to all animals [10].

Following recovery, the cats were moved to their individual cages and received buprenorphine at a dose of 0.02 mg/kg IM and tramadol (Tramal; Vianex, Athens, Greece) at a dose of 1 mg/kg SC BID for 5 days [10]. The pain was evaluated by using the Colorado State University Feline Acute Pain Scale, while the cats were alert, before and after the bandage changes. If a cat was in pain for more than 5 days, based on daily physical examination, tramadol was administered until signs of pain resolved. Pethidine, at a dose of 3 mg/kg IM (Pethidine Hydrochloride; Monico SPA, Venice, Italy), was used for rescue analgesia if the pain score was ≥3. 

### 2.5. Visual Observations

The same investigator observed and recorded the open wounds at each bandage change. The time it took until the granulation tissue initially appeared and filled the entire defect, the amount of fluid within the bandage, and any indications of infection or additional abnormalities were all noted. Bandages were changed daily for the next 25 days using dexmedetomidine (DexDomitor; Orion Corporation, Espoo Finland) at a dose of 0.04 mg/kg IM. Reversal of sedation was achieved with the administration of Atipamezole (Alzane; Zoetis, Athens, Greece) [10]. Clinical evaluation was performed by visual observation daily for 25 days while the cats were sedated, using the previously described protocol, with dexmedetomidine and performed by the same clinician, who was not aware of the treatment provided, using a subjective wound evaluation sheet. It included a visual inspection of the wounds based on other references [1,10]. Signs of infection, including discharge or pus from the wounds, were noted by the clinician. Suspect wounds underwent aerobic culture and sensitivity testing, as per institutional standard pathology laboratory protocol. The bandages were changed under aseptic conditions. After the bandage removal and before treatment application, wounds and surrounding skin were carefully cleaned using sterile saline-soaked gauze to avoid wound-bed disturbance. All the other measurements including planimetry, laser Doppler flowmetry, and histologic examination were made on days 0, 7, 14, and 25 under general anesthesia with the protocol described above [10].

### 2.6. Laser Doppler Flowmetry (LDF)

For measuring tissue perfusion, the animals were acclimatized to the study room (temperature 20–22 °C) and a laser Doppler velocimeter was used (Laserflo BPM2, Vasomedics, St. Paul, MN, USA). 

Measurements were taken on days 0, 7, 14, and 25, with the probe placed perpendicular to the wound. Each time, 3 separate readings at 5 s intervals were recorded and a mean value was obtained. After each measurement, an alcoholic solution (Hartmann Manusept Sterillium gel, Heidenheim, Germany) was applied to the probe for asepsis. Following drying, the probe was cleaned with a dry sterile gauze before the next measurement. Values were expressed in perfusion units (mm/s).

### 2.7. Planimetry

Planimetry was performed on days 0, 7, 14, and 25 following LDF. Digital images of the defects were taken using a digital camera with a ruler (mm) in the image frame. Also, a sterile ruler was placed at the edges of the defect. The images were uploaded into a wound-tracing software program (NIH ImageJ; http://rsb.info.nih.gov/ij/index.html accessed on 19 June 2022). Measurements of these surfaces were performed blindly by the same clinician who was not aware of the exact wound care product that was applied.

The % epithelization, % wound contraction, and % total wound healing were measured on days 7, 14, and 25 based on the formulas reported by Bohling et al. as follows [2]:% epithelization = area of epithelium day n/total wound area day n × 100;% contraction = 100 – Χ,

where Χ = total wound area day n/original wound area day 0 × 100;

% total wound healing n = 100 − Υ,

where Υ = open wound area day n/original wound area day 0 × 100 [1,39,40].

### 2.8. Histologic Evaluation

Tissue samples were obtained from all wounds from different locations each time (corners of the square wound) using a 4 mm biopsy punch for histologic assessment of the 4 wounds, following the tissue perfusion measurements on days 0, 7, 14, and 25. The excised skin and underlying tissue served as a day 0 biopsy. Specimens were anonymized so that the 2 evaluating pathologists were not aware of the group. Both pathologists reviewed all specimens using a scoring system based on Gillette et al. [41]. All biopsies were routinely processed, paraffin-embedded, sectioned at 4–5 μm, and stained with hematoxylin and eosin. The inflammatory cell infiltration, edema, angiogenesis, fibroblast concentration, and collagen production scores were evaluated semiquantitatively in 1 high-power field (HPF 400× magnification) per section, using the following scoring system based on Gillette et al. [41]. The assessment of inflammatory cell infiltration was conducted by determining the quantity of infiltrating neutrophils, eosinophils, lymphocytes, plasma cells, macrophages, and mast cells per high-power field (HPF) as follows: <3 cells/HPF was considered normal (0), 3–10 cells/HPF was considered mild (1), 11–30 cells/HPF was considered moderate (2), and >31 cells/HPF was considered to be a marked infiltration (3). The edema was scored as normal (0), mild (1), moderate (2), and marked (3) i depending on the degree of separation of cells and collagen by acellular material. To be more specific, slight separation was categorized as mild edema (1), a separation of 30–50 μm as moderate edema (2), and a separation of >50 μm as substantial edema (3). The following criteria were used to determine the angiogenesis score: 0: <3 vessels; 1: 3–10 vessels; 2: 11 to 30 vessels; or 3: ≥31 vessels per HPF. The fibroblast concentration was assigned a score of 0 for fibroblasts organized in a random pattern, typical of normal debris; 1 for a mild increase in localization of fibroblasts in the wound tissue in numbers of 3–10 per higher-power field; 2 for a moderate increase in localization of fibroblasts in the wound tissue in numbers of 11 to 30; and 3 for a marked increase in localization of fibroblasts in the wound tissue in numbers ≥ 31 per HPF. The collagen density score was considered as 0 for no collagen production, 1 for slight separation of fibroblasts by collagen bundles, 2 for a dense accumulation of collagen between fibroblasts, and 3 for extensive separation of fibroblasts by collagen.

### 2.9. Statistical Analysis

The data were checked for normality using the Shapiro–Wilk W test. All statistical analyses were conducted using computer software (SPSS Statistical Package for the Social Sciences, version 25.0, with the year of release 2017). For parametric tests, data were expressed as mean ± SD. For non-parametric tests, data were expressed as median (range). For the variables percentage of epithelialization, contraction, total wound healing, and LDF, double-repeated measures analysis of variance was used. These variables were the dependent variables, and the sample unit consisted of eight cats, with the independent variables being the number of wounds (4 wounds) and the sampling day (0, 7, 14, 25 days) [42]. The non-parametric statistical test of Wilcoxon was performed for the variables, cell infiltration, edema score, collagen production, fibroblast concentration, and angiogenesis [42]. The degrees from 0 to 3 were recoded into 1 to 4 to estimate descriptive statistics. The Monte Carlo simulation method was used to estimate the observed significance levels (p-values) of statistical tests, that were predetermined at α = 0.05 (*p* ≤ 0.05) [43].

The sample size was determined with the G*Power software (ver. 3.1). The determination of the sample size was performed using a patient-based analysis. The angiogenesis, edema, and collagen production scores were evaluated. Eight cats were required if there was (1) a difference of 5.07 blood cells in high-power fields (HPFs) (± 0.4 blood cells in HPF SD within each group) in the number of new cells (angiogenesis) between the groups; (2) 0.1 inflammatory cells were detected per HPF (± 0.27 blood cells in HPF SD within each group); and (3) 0.37 cells in collagen production in HPF (± 0.95 collagen increase in HPF SD within each group) were detected at a significance level of alpha = 0.05 with a power of (1-beta) = 0.80 using a conservative F test—ANOVA testing approach for repeated measures, within–between interaction. The estimation of a common SD was based on the study of Karayannopoulou et al. (2015) [6]. 

## 3. Results

### 3.1. Visual Observations

The mean time to the first appearance of granulation tissue for the MGH-treated wounds was 5.13 ± 1.31 days while for the HP-treated wounds it was 4.25 ± 1.25 days. The mean time to fill the entire wound with granulation tissue in MGH-treated wounds was 14.25 ± 2.51 days, while for HP-treated wounds it was 12.63 ± 1.31 days. The mean time to the first appearance of granulation tissue was 5 ± 1.07 days for the HC and 4.63 ± 1.30 days for the HPC. The mean time to fill the entire wound with granulation tissue was 14.13 ± 1.89 days for the HC and 12.5 ± 1.20 days for the HPC. Serosanguinous drainage was observed in all wounds; however, the volume of the fluid was not measured. 

All cats recovered uneventfully from the anesthesia. No feline patient had a pain score over 2 using the Colorado State University Feline Acute Pain Scale, so rescue analgesia was not administered. All cats, post-surgically, ate ad libitum, maintaining their body weights, and eventually were given for adoption. 

### 3.2. Laser Doppler Flowmetry (LDF)

Tissue perfusion increased significantly on days 7 and 14 compared to day 0 and then decreased significantly on day 25 compared to day 0 in both MGH- and HP-treated wounds (*p* = 0.003). MGH-treated wounds (2.14 ± 0.18 mm/s) and HP-treated wounds (2.02 ± 0.13 mm/s) had significantly greater tissue perfusion than the untreated controls (HC: 1.59 ± 0.11 mm/s; HPC: 1.60 ± 0.05 mm/s) (*p* = 0.001). The mean values of LDF measurements did not differ significantly between the MGH-treated and HP-treated wounds. All LDF measurements are presented in Table 1.

### 3.3. Planimetry

During the healing process, the % epithelialization, % wound contraction, and % total wound healing increased significantly in all wounds (*p* < 0.001). No significant differences were observed between the treatments and the untreated control wounds (MGH and HC, HP and HPC) or between the different treatments (MGH and HP). The epithelialization, contraction, and total wound healing measurements are presented in Table 2.

No full epithelialization was reached in our study. All figures of wound healing at all measurement times are included in Figure 2 and Figure 3.

### 3.4. Histologic Evaluation

All wounds were compared histologically including MGH–HC, HP–HPC, and MGH–HP groups. No significant differences in the inflammatory cell infiltration score were found between the treated wounds at any measured time point. The median edema score was lower in the MGH-treated wounds (2; range 1–4) compared to the HC-treated wounds (3; range 2–4) on day 7 (*p* < 0.05) (Figure 4). 

The median angiogenesis score was higher on day 7 in the MGH-treated (2; range 1–3) compared to the HP-treated wounds (2; range 1–2) (*p* = 0.046) (Figure 5). 

On day 25, the fibroblast concentration score was increased in the MGH-treated wounds (3.5; range 3–4) compared to the HP-treated wounds (3; range 2–4) (*p* = 0.046) (Figure 6). 

There were no differences concerning the collagen production scores between the MGH- and HP-treated wounds at any measured time point. No significant differences in inflammatory cell infiltration, edema, angiogenesis, fibroblast concentration, or collagen production scores between treated wounds and untreated controls were observed. The histological patterns of the wounds at all measurement times are included in Figure 7, Figure 8, Figure 9 and Figure 10.

## 4. Discussion

We evaluated the effect of L-Mesitran ointment and Hypermix ointment applied daily for 25 days on second-intention cutaneous wound healing in cats. Planimetry, laser Doppler flowmetry, and daily clinical and histologic examination on days 0, 7, 14, and 25 were used to assess wound healing. To the authors’ knowledge, this is the first study investigating the effect of these products on full-thickness wounds in cats based on visual observation, LDF, planimetry, and histologic evaluation. The study showed that there was a higher score of angiogenesis with MGH-treated wounds compared to HP-treated wounds, but not greater tissue perfusion, as measured using Doppler flow. The MGH- and HP-treated wounds showed greater tissue perfusion than untreated controls. The MGH-treated wounds had a lower edema score than the HC on day 7. Finally, on day 25, the MGH-treated wounds had an increased fibroblast concentration score compared to the HP-treated wounds when tissue samples were examined histologically. No significant differences in inflammatory cell infiltration, edema, angiogenesis, fibroblast concentration, or collagen production scores between the treated wounds and untreated controls were found. There was no difference in clinical wound parameters. Overall, in the study presented here, medical-grade honey or Hypericum did not promote wound healing in acute full-thickness cutaneous defects in cats compared to their untreated controls. 

We found significant differences between the treatment and control groups in terms of LDF measurements. Specifically, tissue perfusion was significantly higher in HP- and MGH-treated wounds compared to untreated control wounds. The relationship between tissue perfusion and wound healing has been well described in dogs and cats [5,6,7,8,9,10]. According to other studies, cutaneous microcirculation favors granulation tissue production and cutaneous healing by delivering nutrients and oxygen to the wound site [6,39]. Furthermore, in our study, tissue perfusion increased significantly on days 7 and 14 and then declined by day 25. The same phenomenon was observed in the Bohling et al. study, in which tissue perfusion decreased in feline and canine wounds after day 14 [40]. Our findings from LDF measurements in MGH-treated wounds were in agreement with the increased angiogenesis that was found histologically in the MGH-treated wounds on day 7. Based on our findings, it seems that both the MGH and HP treatments induced a significant increase in tissue perfusion in cutaneous wound healing in cats. The increase in tissue perfusion observed in MGH-treated wounds may be due to the vasodilatory effect of honey in the existing blood vessels of the wounds [12]. Increased angiogenesis observed in MGH-treated wounds in the present study constitutes an additional factor that could evoke an increase in tissue perfusion. Contrarily, no significant difference in angiogenesis was revealed between the HP-treated and control groups. However, the inflammatory response also implies increased blood supply to the area. The relatively intense inflammatory phase of HP-treated wounds compared to the MGH-treated wounds may explain why the former showed increased perfusion. Nevertheless, since this difference was not statistically significant, further investigation is needed.

No significant differences were found in epithelialization, wound contraction, and wound healing between the treated wounds and untreated controls nor between the treatments themselves. Perhaps these were ideal conditions, with daily aseptic bandage changes, as there was no evidence of wound infection, and these products might show different actions in an infected environment. A recent study in dogs reported that epithelialization was about 10% higher at days 16, 18, and 21 following treatment of acute full-thickness wounds with a product containing manuka honey compared to standard-of-care (CON) dressings, although wound contraction and histological scores did not differ between manuka honey and standard-of-care dressings in that study [33]. In another recent study in dogs, the percentage of epithelialization was higher in acute full-thickness wounds treated with manuka honey and calcium alginate dressings compared to wounds treated with calcium alginate dressings alone [21]. The healing of full-thickness wounds in cats is different from in dogs, as epithelialization proceeds slower in feline than in canine patients [1]. It seems that cutaneous wound healing in cats is not enhanced by the local application of MGH or HP and no difference was observed between MGH and HP regarding wound-healing enhancement. 

The main difference between “real-life” clinical findings and the current study might be the cleanness of the wounds. Full-thickness wounds created in an in vivo model under anesthesia and in clean conditions, and with the animals being housed in a clean laboratory setting, are typically not colonized with bacteria and less prone to infection, in contrast to chronic wounds that are often infected and do not heal. 

Several other clinical studies have shown that L-Mesitran promoted the healing of different types of (infected) wounds while diverse previous treatments failed, suggesting a superior activity, and reiterating its cost-efficacy [13,44,45,46,47,48]. According to a series of 10 cats with cutaneous contaminated wounds that were treated with L-Mesitran, regrowth of hair and minimal scarring were evident [8]. In addition, medical-grade honey was applied successfully on a degloving injury in a cat that eventually healed by second intention [9].

Based on our findings, MGH-treated wounds showed a decreased edema score compared to HC wounds, possibly because during the repair stage of healing at day 7, the osmotic effect of the honey contributed to a decrease in edema formation [11]. We found that there were significant differences in the angiogenesis scores between the two treatment groups. More specifically, angiogenesis was significantly higher on day 7 in MGH-treated wounds compared to HP-treated wounds. This finding might be related to the increased tissue perfusion of the MGH-treated wounds that was observed in our study. There have been a few in vitro studies published describing its prο-angiogenic effect [49,50]. Medical-grade honey has been shown to stimulate angiogenesis in humans and buffaloes [50,51]. This may be due to the generation of hydrogen peroxide, low levels of which may stimulate angiogenesis [52]. According to a study, hyperforin, a key ingredient of Hypermix, inhibits angiogenesis in the chorioallantoic membrane assay in vivo and in vitro in normal bovine aortic endothelial cells [53]. This contrasts with the results of a recent study that have shown that certain angiogenesis-promoting factors are enhanced in colorectal micro-tumors formed on the chorioallantoic membrane by hyperforin and photodynamic therapy with hypericin [54]. This contrasts with the angiogenic effects of honey, as was shown in our study. 

In the present study, we found that the application of medical-grade honey on MGH-treated wounds increased the fibroblast concentration score (on day 25) in contrast to HP-treated wounds. Fibroblasts play a crucial role in the wound-repairing process, from the late inflammatory phase until the occurrence of full final epithelization of the injured tissue [55,56,57]. Wound fibroblasts aid in granulation tissue formation and play a role in the elaboration, orientation, and contraction of the wound extracellular matrix into connective tissue [57]. The increased fibroblast concentration seen in our study may reflect increased scarring, which could be a negative finding [58].

Limitations of the study include the low number of cats, although a power analysis was performed. A higher number of cats would produce differences between treatment and control groups regarding epithelialization, contraction, total wound healing inflammatory cell infiltration, edema, angiogenesis, fibroblast concentration, and collagen production scores. Clean surgical wounds in healthy cats supported a favorable outcome in second-intention healing, whereas management of such wounds in clinical cases may influence the outcome of the healing process. Results might be different in clinical situations, in chronic or infected wounds, or if an objective mechanism for monitoring visual changes in the wounds is utilized. Moreover, the usage of daily sedation with a vasoconstrictive agent (dexmedetomidine) might have an impact on real-time tissue perfusion measurements, although the same sedative agent was used in all animals and so all wounds were equally affected. The results might differ if the tacking sutures were not placed in the wound corners, treatments and controls were placed on the same side of the animal, or the control samples included the base ointment without the active ingredient. 

## 5. Conclusions

Daily topical application of MGH or HP for 25 days did not accelerate the healing process of acute cutaneous wounds in cats. The results did not translate to a clinically meaningful difference in wound healing in this model. Further investigations are needed in a clinical setting including chronic or infected wounds to determine the benefit of either MGH or HP on feline wound healing.

## Figures and Tables

**Figure 1 animals-14-00036-f001:**
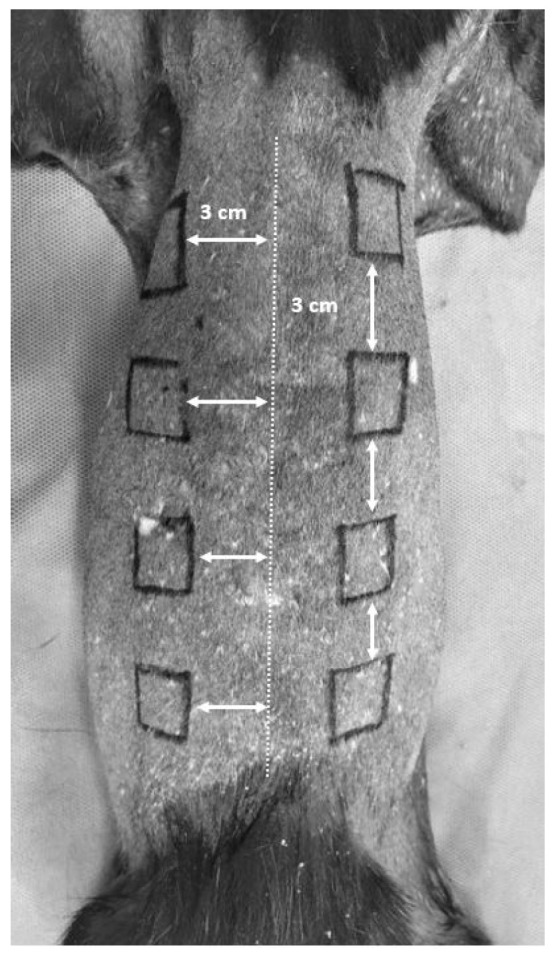
Eight 2 × 2 cm squares were drawn, four on either side of the dorsal midline. The squares were 3 cm away from each other and 3 cm away from the dorsal midline.

**Figure 2 animals-14-00036-f002:**
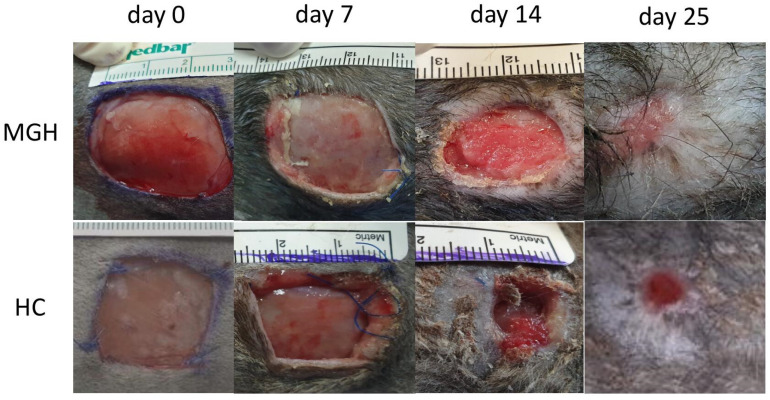
MGH-treated and HC wounds at different measurement times (days 0, 7, 14, 25).

**Figure 3 animals-14-00036-f003:**
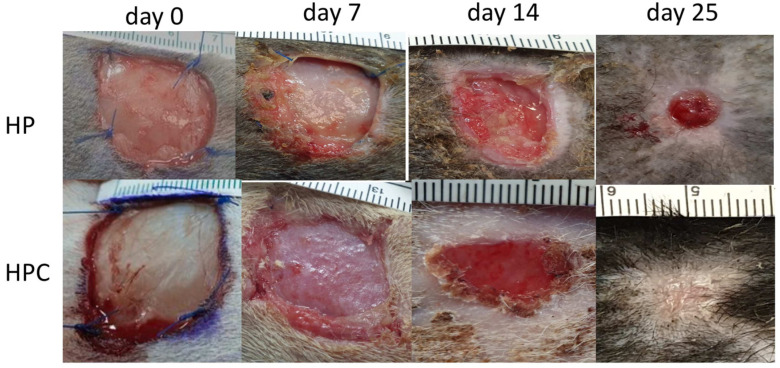
HP-treated and HPC wounds at different measurement times (days 0, 7, 14, 25).

**Figure 4 animals-14-00036-f004:**
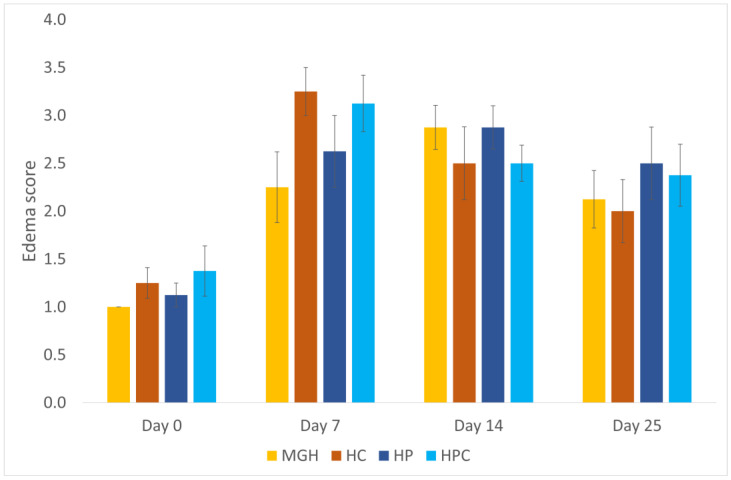
Edema score of honey-treated wounds (MGH) compared to Hypericum-treated wounds (HP), and their controls (HC and HPC). Honey-treated wounds showed a significantly lower edema score than HC on day 7. Bars represent the mean ± standard deviation (yellow = MGH, orange = HC, dark blue = HP, light blue = HPC).

**Figure 5 animals-14-00036-f005:**
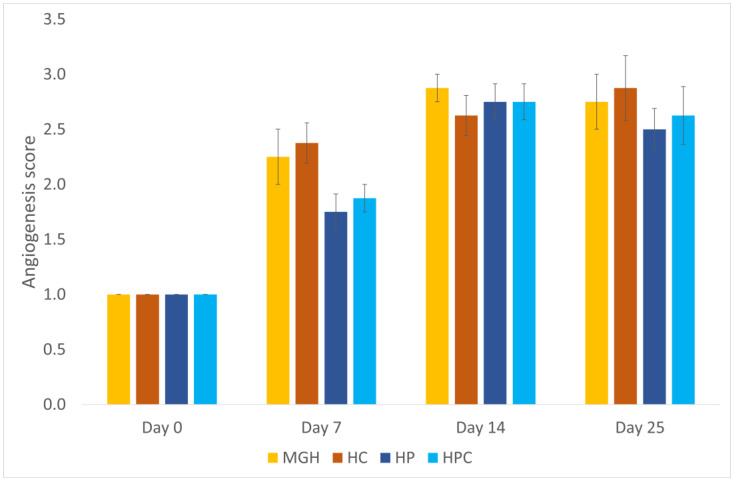
Mean angiogenesis score of honey-treated wounds (MGH) compared to Hypericum-treated wounds (HP), and their controls (HC and HPC). Honey-treated wounds showed a significantly greater score than HP-treated wounds on day 7. Bars represent the mean ± standard deviation (yellow = MGH, orange = HC, dark blue = HP, light blue = HPC).

**Figure 6 animals-14-00036-f006:**
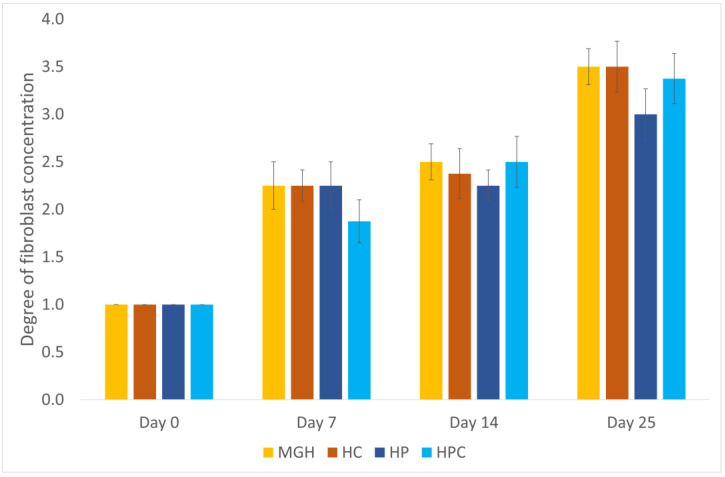
Mean fibroblast concentration scores of honey-treated wounds (MGH) compared to Hypericum-treated wounds (HP) to their control wounds (HC and HPC). Honey-treated wounds had significantly increased fibroblast concentration scores compared to HP-treated wounds on day 25. Bars represent the mean ± standard deviation (yellow = MGH, orange = HC, dark blue = HP, light blue = HPC).

**Figure 7 animals-14-00036-f007:**
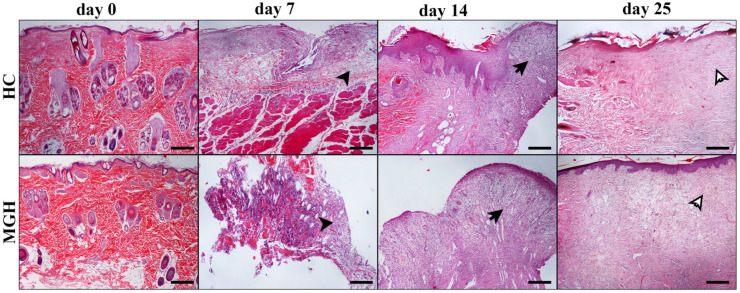
Representative images of tissue sections from honey-treated (MGH) wounds and their controls (HC) at different measurement times (days 0, 7, 14, 25): On day 0, the excised skin in both images appears normal. On day 7, the edema of the HC wound compared to the MGH-treated wound is more intense, showing prominent separation of cells and collagen by acellular material (black arrowheads). On day 14, no significant differences are depicted between the two groups concerning the degrees of inflammatory cell infiltration, edema, fibroblast, and blood vessel aggregation, as well as collagen density (black arrows). Finally, on day 25, in MGH-treated wounds, a little lower collagen density and fibroblast accumulation and relatively increased capillary aggregation, relative to HC wounds, were noticed (white arrows). The formation of the epidermis is characteristic in both images, although in MGH it seems closer to the normal architecture. MGH and HC/days 0, 7, 14, 25: H-E; Hematoxylin and eosin (H&E) staining, magnification 100×, bar = 100 μm.

**Figure 8 animals-14-00036-f008:**
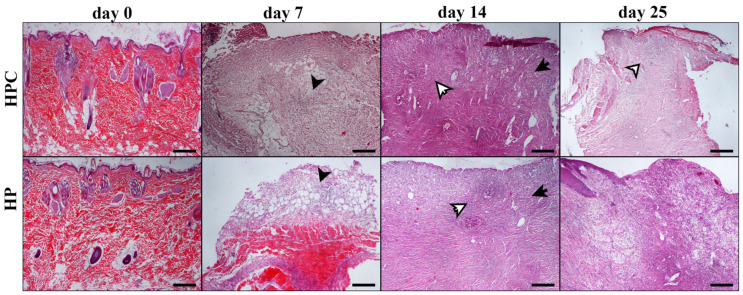
Representative images of tissue sections from hypericum-treated (HP) and HPC wounds at different measurement times (days 0, 7, 14, 25). On day 0, images show normal skin histology. On day 7, both HPC and HP-treated wounds present inflammatory cell infiltration and edema, with the former showing a little higher intensity than the latter (black arrowheads). The opposite is noticed regarding fibroblast aggregation and collagen density. On day 14, mild edema is shown in both groups, accompanied by pronounced fibroblast aggregation of a little higher intensity in HP-treated wounds relative to HPC (black arrows). The opposite is observed for inflammatory cell infiltration (white arrows). Finally, on day 25, intense collagen density and fibroblast aggregation is noticed in the HPC wound (white arrowhead). The wounds exhibit incomplete epithelial coverage. HPC and HP/days 0, 7, 14, 25: H-E; Hematoxylin and eosin (H&E) staining, magnification 100×, bar = 100 μm.

**Figure 9 animals-14-00036-f009:**
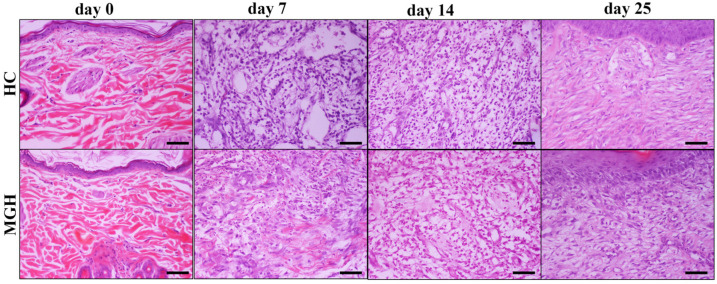
Representative images of tissue sections from honey-treated (MGH) wounds and their controls (HC) at different measurement times (days 0, 7, 14, 25): On day 0, the excised skin in both images appears normal. On day 7, the edema of the HC wound compared to the MGH-treated wound is more intense, showing prominent separation of cells and collagen by acellular material. On day 14, no significant differences are depicted between the two groups concerning the degrees of inflammatory cell infiltration, edema, fibroblast, and blood vessel aggregation, as well as collagen density. Finally, on day 25, in MGH-treated wounds a little lower collagen density and fibroblast accumulation and relatively increased capillary aggregation, relative to HC wounds were noticed. MGH and HC/days 0, 7, 14, 25: H-E; Hematoxylin and eosin (H&E) staining, magnification 200×, bar = 50 μm.

**Figure 10 animals-14-00036-f010:**
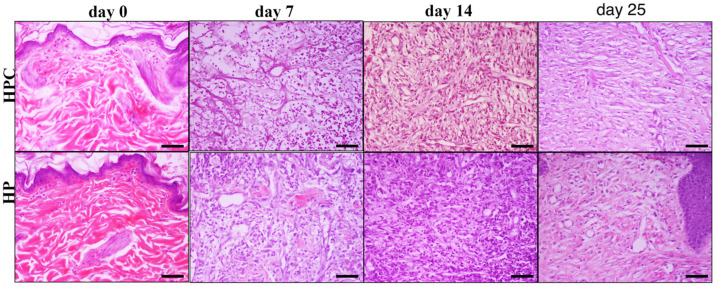
Representative images of tissue sections from hypericum-treated (HP) and HPC wounds at different measurement times (days 0, 7, 14, 25). On day 0, images show normal skin histology. On day 7 both HPC and HP-treated wounds, present inflammatory cell infiltration and edema, with the former showing a little higher intensity than the latter. The opposite is noticed regarding fibroblast aggregation and collagen density. On day 14, mild edema is shown in both groups, accompanied by pronounced fibroblast aggregation of a little higher intensity in HP-treated wounds related to HPC. The opposite is observed for inflammatory cell infiltration. Finally, on day 25, increased collagen density and fibroblast aggregation is mostly noticed in the HPC-treated wound. HPC and HP/days 0, 7, 14, 25: H-E; Hematoxylin and eosin (H&E) staining, magnification 200×, bar = 50 μm.

**Table 1 animals-14-00036-t001:** Tissue perfusion measurements using LDF in MGH- and HP-treated wounds and controls (HC and HPC) in 8 cats. Data are expressed as mean ± SD. Values are expressed in mm/s.

Groups	Tissue Perfusion in Full-Thickness Wounds			
	Day 0	Day 7	Day 14	Day 25
MGH	1.85 (±0.59)	2.90 (±0.82) *	2.26 (±0.80) *	1.54 (±0.95) *
HC	1.75 (±0.41)	1.98 (±0.51) *	1.59 (±0.56) *	1.06 (±0.41) *
HP	1.71 (±0.48)	2.36 (±0.68) *	2.24 (±0.92) *	1.69 (±0.78) *
HPC	1.66 (±0.43)	1.80 (±0.37) *	1.87 (±0.70) *	1.07 (±0.77) *

* Statistically significant differences between groups or between consecutive measurement times in the same group.

**Table 2 animals-14-00036-t002:** Photoplanimetry measurements in MGH- and HP-treated wounds and control (HC and HPC) in 8 cats (data are expressed as mean ± SD).

Group	Day	Epithelialization %	Contraction %	Total Wound Healing
MGH	0			
	7	13.95 (±5.14)	22.60 (±11.80)	43.11 (±14.18)
	14	50.50 (±11.23)	58.64 (±9.12)	77.54 (±7.56)
	25	86.58 (±14.95)	86.45 (±6.99)	97.86 (±3.00)
HC	0			
	7	11.46 (±4.28)	22.08 (±14.81)	43.76 (±17.60)
	14	47.86 (±6.72)	69.64 (±11.05)	83.43 (±6.06)
	25	88.14 (±10.41)	84.78 (±4.99)	98.41 (±1.44)
HP	0			
	7	12.89 (±4.34)	20.54 (±8.97)	44.35 (±10.05)
	14	56.61 (±8.20)	56.83 (±10.20)	78.68 (±6.73)
	25	79.61 (±14.75)	83.59 (±8.61)	95.31 (±3.82)
HPC	0			
	7	8.38 (±4.90)	21.15 (±11.47)	43.29 (±14.06)
	14	49.63 (±6.54)	60.35 (±11.66)	79.23 (±8.53)
	25	80.1 (±17.91)	81.28 (±6.27)	96.25 (±3.38)

## Data Availability

Data are contained within the article.

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
