# Peer review of "Evaluation of the Effectiveness of Medical-Grade Honey and Hypericum Perforatum Ointment on Second-Intention Healing of Full-Thickness Skin Wounds in Cats"

_animals, 2023, doi:10.3390/ani14010036_

Round 1

Reviewer 1 Report

Comments and Suggestions for Authors

Overall this is a great study and with valuable findings in wound healing for cats. I do agree with some of your comments that the clinical applicability is not completely accurate. If no sutures were placed that could have significantly affected the wound/remodeling/contraction phase of healing, but you had controls to compare this against. Additionally, most wounds are commonly infected which I suspect would have had significantly different results as we know MGH has good antibacterial properties and the control would have worsened.

The only comment that I would recommend changing are the colors on the bar graphs for Figures 5 and 6. If the reader looked at Figure 4 they would assume the following bar graphs may also be blue for Honey and orange for Control. Potentially change the orange on Figure 5 and 6 since those are both for Hypericum. Additionally, you could make the key and words slightly larger if possible as they are tough to read. 

Reviewer 2 Report

Comments and Suggestions for Authors

Comment 1

Line 25 – I recommend replacing this repetition “the effectiveness of medical- grade honey and Hypericum “ with “their effectiveness”.

Comment 2

Line 71-76 – I recommend using this info in the discussion part.

Comment 3

Line 122 – is it Petrolatum?

Comment 4

Line 531 – Were these taking sutures placed in all the defects created? From the two seriated pictures I saw in the results section the sutures can be seen only in one. Could you please mention their exact role in your study? Wouldn’t these sutures prevent the normal contraction process of these wounds?

Comment 5

Discussion – your partial conclusions discussed in this section are a bit confusing because you say in some paragraphs that there are positive differences in some moments of the treatment but afterwards you conclude that no significant positive effects were observed. You should be clearer in comparing the groups and the parameters evaluated between groups or try organizing better the information.

Comments on the Quality of English Language

I also recommend English revision for the first part of the manuscript especially the simple summary and the introduction

Reviewer 3 Report

Comments and Suggestions for Authors

In this paper, the authors test the efficacy of medical grade honey and Hypericum on full-thickness, excisional wound healing in cats. Wound closure was measured by planimetry, and angiogenesis was measured by laser-doppler flowmetry. All other results were based on visualizing hematoxylin and eosin-stained skin sections. This is the major flaw of the paper because collagen production, fibroblast proliferation and levels of edema cannot be measured accurately by this method.

The overall conclusion is that neither medical grade honey nor Hpericaum accelerated wound closure. Wound healing is a nuanced process and healing too quickly can lead to scar formation which is more fragile that non-scarred tissue. It would be beneficial to look more closely at their wound biopsies to determine if there are differences in collagen deposition (using an actual collagen stain). 

Major Concerns:

It is not clear what the difference is between the “HC” and “HPC” groups. If these are both untreated controls, then they should be combined as one group in Tables 1 & 2.

Do you only have planimetry for days 0, 7, 14 and 25? What about the other days? If you have that data, graph as a line graph. Include untreated controls (combined), MGH and HP data.

Combine Figures 2 & 3. Organize and label the photos as in Figure 7.

Add untreated control images to the combined Figure 2/3. 

Figure 4. Include HP data. Describe bars in legend (blue = honey, orange = control). Make fonts for x and y axes bigger

Figure 5 Include control wound data. Describe bars in legend (blue = honey, orange = control). Make fonts for x and y axes bigger

In all figures change “standard deviation bars were evident” to “bars represent the mean ± standard deviation:

Figure 6. Include control wound data. Describe bars in legend (blue = honey, orange = control). Make fonts for x and y axes bigger

Rewrite results for the updated Figures 2 - 6

Explain how fibroblasts were identified. Did you perform immunostaining?

Explain how collagen production was measured. Did you perform Trichrome staining? Did you perform PCR or western blot?

Figure 7. Are these photos all taken with the same magnification? Please add a scale bar to each photo.

Figure 8. Are these photos all taken with the same magnification? Please add a scale bar to each photo.

If you intend to use the photos in Figures 7 & 8 for edema, collagen & fibroblast data, then you need to show enlarged images with arrows pointing to edema, fibroblasts and collagen OR adjust the results/discussion to not over-interpret this data.

In the discussion, the authors state “on day 25 MGH wounds had an increased fibroblast proliferation score than the HP wounds” This statement needs to be removed because they did not perform any method (such as Ki67 /fibroblast double immune-fluorescence staining) to assess proliferation. 

Minor comments:

In the simple summary two sentences repeat the same information. Remove the repeated sentences (in italics):

Tissue perfusion was better in medical grade- honey and Hypericum wounds rather than on their un-treated controls. Medical grade- honey wounds showed lower edema, higher angiogenesis and increased fibroblast proliferation compared to Hypericum wounds. Medical-grade honey and Hypericum increased tissue perfusion compared to their untreated controls. Medical- grade honey wounds had histologic parameters superior to Hypericum in terms of angiogenesis and fibroblast proliferation in cutaneous wound healing in cats.

The methods discuss evaluation of inflammatory cell infiltration, but no data was shown. This should be removed from the methods section.

Round 2

Reviewer 3 Report

Comments and Suggestions for Authors

The authors re-graphed Figures 4-6 which make the data much easier to interpret. They also added Figures 9 and 10 which show the H&E stained wound sections at a higher magnification. This is very helpful and lends greater support to their conclusions

Minor comments:
Figure 7 legend: What does this sentence mean? "The control section bares intact (left half) and wounded area (right half). "

Is magnification really 10X for figures 7 & 8? Shouldn’t it be 100X?

Is magnification really 20X for figures 9 & 10? Shouldn’t it be 100X?

“Barr” should be written as “bar"

Author Response

Figure 7 legend: What does this sentence means? "The control section bares intact (left half) and wounded area (right half)"

Response: This sentence was deleted.

Is magnification really 10x for Figures 7&8? Shouldn't it be 100x?

Response: The authors wrote the magnification of the objective lens. However we changed it with total magnification. Since we added bars, magnification could also be deleted. 

Is magnification really 20x for Figures 9&10? Shouldn't it be 200x?

Response: The authors wrote the magnification of the objective lens. However we changed it with total magnification. Since we added bars, magnification could also be deleted. 

"Barr" should be written as "bar"

Response: The word "barr" was corrected.